# Structural Antibiotic Surveillance and Stewardship via Indication-Linked Quality Indicators: Pilot in Dutch Primary Care

**DOI:** 10.3390/antibiotics9100670

**Published:** 2020-10-03

**Authors:** Alike W. van der Velden, Mieke I. van Triest, Annelot F. Schoffelen, Theo J. M. Verheij

**Affiliations:** 1Julius Center for Health Sciences and Primary Care, University Medical Center Utrecht, Universiteitsweg 100, 3584CG Utrecht, The Netherlands; t.j.m.verheij@umcutrecht.nl; 2National Institute for Public Health and the Environment, Antonie van Leeuwenhoeklaan 9, 3721MA Bilthoven, The Netherlands; m.i.vantriest-2@prinsesmaximacentrum.nl (M.I.v.T.); Annelot.schoffelen@rivm.nl (A.F.S.)

**Keywords:** antibiotic, general practice, quality indicator, prescribing quality, feedback, national surveillance, respiratory tract infection, episode, clinical indication

## Abstract

Insight into antibiotic prescribing quality is key to general practitioners (GPs) to improve their prescribing behavior and to national antibiotic surveillance and stewardship programs. Additionally to numbers of prescribed antibiotics, quality indicators (QIs) linked to the clinical indication for prescribing are urgently needed. The aim of this proof of concept study was to define indication-linked QIs which can be easily implemented in Dutch primary care by collaborating with data-extraction/processing companies that routinely process patient data for GP practices. An expert group of academic and practicing GPs defined indication-linked QIs for which outcomes can be derived from routine care data. QI outcomes were calculated and fed back to GPs from 44 practices, associations between QI outcomes were determined, and GPs’ opinions and suggestions with respect to the new set were captured using an online questionnaire. The new set comprises: (1) total number of prescribed antibiotics per 1000 registered patients and percentages of generally non-1st choice antibiotics; (2) prescribing percentages for episodes of upper and lower respiratory tract infection; (3) 1st choice prescribing for episodes of tonsillitis, pneumonia and cystitis in women. Large inter-practice variation in QI outcomes was found. The validity of the QI outcomes was confirmed by associations that were expected. The new set was highly appreciated by GPs and additional QIs were suggested. We conclude that it proved feasible to provide GPs with informative, indication-linked feedback of their antibiotic prescribing quality by collaborating with established data extraction/processing companies. Based on GPs’ suggestions the set will be refined and extended and used in the near future as yearly feedback with benchmarking for GPs and for national surveillance and stewardship purposes.

## 1. Introduction

General practitioners (GPs) are responsible for prescribing 80% of antibiotics in the outpatient setting and therefore have a major responsibility to continuously focus on appropriate antibiotic prescribing [1]. Inappropriate antibiotic prescribing, defined as non-indicated and/or non-1st choice prescribing, is the main driver of antimicrobial resistance (AMR) [2,3]. AMR has become a major and challenging global health issue [4,5], as it results in reduced efficiency of antibiotics, making treatment of infectious diseases complicated, costly and ultimately ineffective [6]. The reasons for inappropriate antibiotic prescribing are multitude and originate at the interplay between doctor, patient and cultural context. Qualitative research identified many factors driving inappropriate prescribing, like diagnostic uncertainty, lack of guidelines, patient pressure and wrong expectations, time constraints, GPs’ habits, and an unstable GP-patient relationship due to lack of continuity of care [7,8].

To address AMR, antibiotic stewardship programs have been implemented globally. The Center for Disease Control and Prevention defined outpatient antibiotic stewardship as the efforts: (1) to measure antibiotic prescribing; (2) to improve antibiotic prescribing by clinicians and use by patients so that antibiotics are only prescribed and used when needed; (3) to minimize mis- or delayed diagnoses leading to underuse of antibiotics; (4) to ensure that the right drug, dose, and duration are selected when an antibiotic is needed [6].

To support these efforts and for national surveillance purposes, there is an urgent need for a valid instrument to assess antibiotic prescribing quality at a GP practice level. Reporting numbers of prescribed antibiotics does not tell the full story [9,10,11]. This number needs to be related to a valid measure of registered patients or contacts [12], and if done properly, is still not informative for which indications antibiotics may be inappropriately prescribed. Prospective registration of consultations can provide such specific feedback [11,13], and this feedback was shown to sustainably improve antibiotic prescribing behavior [14]. However, regular monitoring by prospective registration will not be accepted by GPs, nor is it feasible for national surveillance purposes. Therefore, a demanding urge exists for antibiotic prescribing quality indicators (QIs) linked to clinical indications, that are informative with respect to over- and non-1st choice prescribing and which can be derived from routine care data [13,15,16,17].

Commissioned by the Dutch Ministry of Health, we aimed to define such a set of antibiotic prescribing QIs, with the following features and prerequisites:(1)Linking to clinical indications, as this was also strongly requested by practicing GPs and the Dutch College of GPs;(2)Outcomes derivable from practices’ electronic patient information systems, based on a uniform data extraction/processing procedure, and done by specialized companies;(3)Appreciated by GPs, as the yearly feedback should serve as a trigger and motivation for targeted quality improvement [14,18]; and(4)Useful for national antibiotic surveillance and stewardship programs.

The main aim of this study was to define an initial concise set of antibiotic prescribing QIs by an expert group of academic and practicing GPs, and investigators in the field of antibiotic research, based on Dutch guidelines and studies using routine care and registration data [14,19,20]. Additional aims were to operationalize data extraction and processing procedures, capturing outcomes for 40 practices, and to investigate the properties of the QI outcomes and GPs’ opinions of the new set.

## 2. Methods

### 2.1. Definition of the Set of Antibiotic Prescribing QIs

The QIs were defined by an expert group of two academic GPs, one practicing GP, a researcher in the field, and a representative of the National Institute for Public Health and the Environment. The group-based in depth discussion was on: (1) experiences of the Dutch ARTI4 prospective registration [11,14]; (2) analysis of antibiotic prescribing using routine care data [20]; (3) the guidelines of the Dutch College of GPs [19]; and (4) relevance for practicing GPs. First, clinical indications of interest and the overall method for data processing were identified. Second, the individual QIs were defined. Complete agreement for each QI was reached in the discussions. The considerations in this process are described in more detail in the Results section.

### 2.2. Setting

Routinely registered data from patients’ medical files were extracted from 44 GP practices by two Dutch data-extraction/processing companies, Medworq (https://www.medworq.nl/) and INSZO-STIZON (https://www.stizon.nl/inszo/). Extraction and processing of patient data is the core business of these companies as they provide GP practices with patient, management, prescribing and chronic care data. These companies were responsible for data-extraction/processing and calculating QI outcomes based on all data from 2017.

### 2.3. Data Processing

For each practice, three data sets were generated:(1)All antibiotic prescriptions for systemic use (ATC code: J01);(2)The mid-period population of registered patients; and(3)All episodes of respiratory tract infections (RTIs) and urinary tract infections (UTIs) in women, whether or not an antibiotic was prescribed, and which one.

To this aim, the data processing step searched for potential infectious ICPC-R chapter codes: R02, R03, R05, R07-R09, R21, R22, R25, R29, R70-R72, R74-R78, R80-R83, and separately for potential infectious ICPC-U chapter codes: U01-U14, U29, U70-U72 in women [20,21]. When such a code was found, all other potential infectious ICPC R or U codes for that patient within three weeks were combined with the first code in a disease episode [22]. Antibiotics prescribed within this period were captured in the episode. After these three weeks, when a new ICPC R or U code was found, this was considered the start of a new episode. Episodes can contain multiple ICPC codes and were named after the most severe ICPC code registered in the three weeks (Appendix A). For example, when ICPC codes for ‘upper RTI’, ‘cough’ and ‘pneumonia’ were registered, this was named an episode of ‘pneumonia’. Based on this most severe code, RTI episodes were furthermore split in an upper- or lower RTI (Appendix A).

Additional to the QI outcomes, the percentage of total J01 prescriptions that were captured in the R and U episodes was calculated for each practice.

### 2.4. QI Outcome Calculations

General QIs:Total number of J01 prescriptions/1000 registered patients/year;Number of J01CR prescriptions/total J01 × 100%;Number of J01FA prescriptions/total J01 × 100%;Number of J01MA prescriptions/total J01 × 100%; andNumber of J01CR + J01FA + J01MA prescriptions/total J01 × 100%

Prescribing percentages for:6.Upper RTI: episodes of upper RTI with any J01/total upper RTI episodes × 100%; and7.Lower RTI: episodes of lower RTI with any J01/total lower RTI episodes × 100%

1st choice prescribing for:8.Tonsillitis: J01CE in R76 episode/R76 episodes with any J01 × 100%;9.Pneumonia: J01AA or J01CA04 in R81 episode/R81 episodes with any J01 × 100%; and10.1Cystitis women: J01XE or J01XX in U71 episode/U71 episodes with any J01 × 100%

### 2.5. Participants

Part of the obligatory continuing education for GPs is organized by collaborating groups of GP practices with their pharmacist(s) who provide prescribing feedback for the subject of the session. Six of those groups, with a total of 44 practices and 104 GPs, volunteered in participating and organizing a session on ‘prudent antibiotic prescribing’. They used the QI feedback developed in this study. Feedback was provided per practice with reference to the other practices in the group, and to outcomes of all practices. AvdV presented this feedback during the educational sessions. Participating practices, from both rural and urban areas, were from the central and southern parts of the Netherlands.

### 2.6. Online Questionnaires

The day after the educational session an invitation and link for an anonymous online survey was e-mailed to participating GPs to evaluate various aspects of the QI set. A reminder was e-mailed 2 weeks later. Answers were allowed on a 7-point Likert scale with answer options ranging from ‘not at all …’ to ‘very …’ and with answer option 4 as ‘moderate’. The specific wording of the questions and answer options are shown in Table 3. Suggestions and remarks concerning the QIs were allowed in free text.

### 2.7. Analyses

QI outcomes are presented as means, with minimum and maximum values, median and quartiles. Pearson’s correlation coefficients were calculated to determine correlations between the QI outcomes.

Questionnaire results are presented by showing percentages of respondents providing the answer options 5–7 (above moderate), or ‘yes’. Free text answers are summarized.

## 3. Results

### 3.1. Definition of the Pilot Set of QIs: Considerations and Decisions of the Expert Team

-As a first quantitative indicator, the number of prescribed antibiotics per 1000 registered patients per year was chosen, as GPs are familiar with this measure. The focus was on how to define the number of registered patients to obtain a valid and comparable outcome. Consensus was for patients registered at the practice, irrespective of a consultation (for infectious diseases), omitting passers-by, and measured at the first day of month 7. Taking the mid-period population was considered to correct for births, deaths and moving in and out of the practice, and is also used by the WHO [23]. Counting every patient registered during the year would result in an overestimation of the population and thereby derogates the QI1 outcome.-In the Netherlands, amoxicillin/clavulanate, macrolides and quinolones are generally non-1st choice antibiotics for those indications where the most antibiotic are prescribed for: upper and lower RTIs, otitis media, uncomplicated UTIs and most skin infections [19,20]. It was therefore considered relevant to include the percentages of those antibiotics, as well as their sum, as an indication for overall non-1st choice prescribing. As these percentages are calculated over all prescriptions, part will be guideline-indicated, for example for complicated disease, sexually transmitted diseases, and in case of allergies. Nevertheless, these percentages were considered informative in comparing with peer practices, and as the higher these percentages are the more likely it is that these antibiotics are prescribed inappropriately.-Prescribing percentages are often calculated as percentages of contacts for a specific ICPC code in which antibiotics were prescribed [20]. A major drawback of this approach is the highly variable patient consultation behavior. In practices with a low threshold to consult, patients can for example consult three times for sinusitis; if they would receive an antibiotic upon the third encounter, the prescription percentage for sinusitis would be 33%. In case the triage is more strict, and patients consult per severe or prolonged disease and then receive an antibiotic, the prescription percentage would be 100%. This example shows that in comparing prescription percentages based on contacts, differences in consultation behavior and triage policy might interfere. There was consensus to calculate prescription percentages per disease episodes of three weeks, irrespective of how many times a patient consults in that period.-International and Dutch studies emphasize that most antibiotics are prescribed for RTIs and that most overprescribing is for RTIs [11,20]. It was therefore agreed to implement two QIs indicative for prescribing quality for RTIs: the prescription percentage for episodes of upper and lower RTIs. During such an illness episode, various ICPC codes can be registered (disease progression, or follow-up), for example bronchitis developing in pneumonia, or upper RTI in sinusitis. It was agreed to generate illness episodes for upper and lower RTI by retrieving and combining all registered potential infectious ICPC codes within a period of three weeks after an index contact. A decision algorithm was defined to name the episode after the most severe ICPC code. Based on this code, RTI episodes were split in upper or lower RTI (Appendix A).-The expert team assumed that this approach would also tackle diagnostic labeling (misclassification), the phenomenon that GPs tend to justify their prescription by registering a more severe infection, for example ‘pneumonia’ [24]. Whether an episode is coded as cough, bronchitis or pneumonia, they all belong to a lower RTI and the prescription percentage is calculated over all these episodes.-Another aspect of inappropriate prescribing is non-1st choice prescribing. The expert group reached consensus to measure non-1st choice prescribing for three specific clinical indications: tonsillitis, pneumonia and cystitis in women. These indications were chosen as they belong to different types of infection, an upper, a lower RTI and a UTI, and because the Dutch guidelines recommend different antibiotic classes for these indications, a small-spectrum penicillin, amoxicillin or doxycycline, and nitrofurantoin or fosfomycin, respectively [19]. The Dutch College of GPs regularly updates the guidelines also taking resistance rates into account in treatment advice for infectious diseases. The increasing resistance of *Streptococcus pneumoniae* to doxycycline (15%) was the reason to change the 1st choice antibiotic for pneumonia to amoxicillin. In 2017, *Strep. pneumoniae* resistance to amoxicillin was 2% [25]. For the period after the guideline change, it was decided to consider both amoxicillin and doxycycline as appropriate treatments. Resistance of *E. coli* to nitrofurantoin and fosfomycin was respectively 2% and 1% in the Netherlands [25].

### 3.2. Outcomes of the QIs

In total, 44 practices participated in the project. Mid-term practice sizes varied between 1406 and 12,660 registered patients, with a mean of 3866. Table 1, with the QI results for these practices, shows large variation in outcomes, for example QI1 varied between 129 and 484 prescribed antibiotics per 1000 registered patients per year, and QI9 between 39% and 100% 1st choice prescribing for pneumonia.

As a control for (correct) ICPC coding and data-processing into episodes, the percentage of all prescriptions (from QI1) that were captured in the upper- and lower RTI and cystitis in women episodes was determined for each practice. This measure varied between 30% and 62%, with a mean of 43%.

### 3.3. Correlations between QI Outcomes

Inter-QI outcome correlations were determined as a proxy for their validity, and also between the QI outcomes and the percentage of prescriptions captured in the episodes (Table 2). Strong positive correlations were found between QI1 (antibiotics/1000 patients/year) and the prescription percentages for upper and lower RTIs (QI6 and 7). The latter two also strongly correlated. QI5 (total 2nd choice prescribing) was significantly correlated with QI1, and with the prescription percentages for upper and lower RTIs. Negative correlations were found between overall 2nd choice prescribing (QI2, 3 and 5) and 1st choice prescribing for tonsillitis (QI8) and pneumonia (QI9). High prescribing for upper and lower RTIs were correlated with low 1st choice prescribing for pneumonia.

For validity and reliability reasons, it is relevant if QI outcomes correlate with the control for ICPC coding and data-processing. None of the QI outcomes correlated with the percentage of prescriptions captured in the episodes.

### 3.4. GPs’ Opinions and Suggestions with Respect to the QIs

In order to have an impact, the set of QIs needs to be appreciated by GPs, and GPs need to trust the data extraction/processing of their electronic patient files. The online questionnaire results are shown in Table 3. The vast majority of GPs confirmed a need for new antibiotic prescribing QIs and the added value of QIs linked to clinical indications. Over 80% of respondents had trust in the data extraction/processing, regarded this set as useful and recognized their prescribing habits in the outcomes. GPs suggested to add more indication-linked QIs, namely for skin infections (n = 10), complicated UTIs (n = 6), acute otitis media (n = 5), sinusitis (n = 2), and STDs (n = 2).

Nearly 70% of GPs expected this QI feedback to change their antibiotic prescribing behavior, mentioning the following reasons: increased reticence towards 2nd choice antibiotics and prescribing for RTIs, more insight in overprescribing, increased awareness, and necessity to re-evaluate old habits. Reasons mentioned by those who did not expect a changing prescribing behavior were: the feedback indicates prudent prescribing already, and these practice outcomes do not apply to me working in a group practice.

The following relevant suggestions and questions were fed back: the request for individual prescriber feedback, whether outcomes can/need to be corrected for complexity/age of the practice patient population, and that prescriptions for long-term, or preventive antibiotic use disproportionally add to the outcome of QI1.

## 4. Discussion

### 4.1. Summary of Main Results

In this study we have shown the feasibility of using routine primary health care data to determine antibiotic prescribing QI outcomes, with additional novelties of linking to clinical indications and analyzing disease episodes. The pilot set of 10 QIs was highly appreciated by GPs and seen as a stimulus to appropriate prescribing habits. Regular analysis and feedback of these QI outcomes thereby significantly adds to national antibiotic surveillance and stewardship programs and can be implemented by other countries as well [6].

### 4.2. Comparison with Existing Literature

Numerous sets of antibiotic prescribing QIs for the outpatient setting have been described and evaluated in the literature [17,26]. To our knowledge, only a limited number of countries have been able to link antibiotic prescribing to clinical indications [20,27,28,29], and none of these do this linking per practice, in a non-academic environment, using routine care data. Moreover, as we have introduced disease episodes instead of analyzing consultations, comparison with other studies is complicated. An extensive set of antibiotic prescribing QIs has been analyzed in Denmark using prospective registration of consultations [13]. Small-spectrum antibiotic prescribing for tonsillitis was considerably higher in Denmark than in our study. It, however, needs to be mentioned that the 1st choice antibiotic, pheneticillin, was periodically unavailable in Dutch pharmacies in recent years. On the other hand, our study shows better adherence to the 1st choice antibiotic for pneumonia.

Studies with QIs often present acceptable ranges, or target values [13]. We have deliberately chosen not to do this yet. First, not enough data are available, upon having introduced prescribing per episode, to make firm recommendations for acceptable ranges. Second, the QI results will be fed back using the existing structure of continuing education for collaborating practices. We anticipate that the combination of benchmarking by comparison to peer practices, as well as to overall averages and quartiles will serve as a stimulus to improve prescribing habits.

The inter-practice variation in QI outcomes reflects the highly variable antibiotic prescribing quantity and quality shown in many other studies [30,31,32,33]. In the Netherlands, antibiotic prescribing/1000 registered patients/year was shown to vary by a factor of five, which makes it credible that QI outcomes show a similar variation [14].

### 4.3. Strengths and Limitations

The overall strength of this study is a new procedure resulting in a set of antibiotic prescribing QIs providing information in which indications antibiotics may be inappropriately prescribed. This can be of much added value to national antibiotic surveillance and stewardship programs, as outcomes can be obtained for nearly all GP practices. We have shown that outcomes can be derived without involvement or time investment of GPs, using their routine electronic patient health records, by professional data-extraction/processing companies. Our procedure could be adapted by other countries, implementing their indications and antibiotic groups of interest, thereby enabling European, or even worldwide surveillance of antibiotic prescribing quality. The involvement of professional companies seems a prerequisite for national implementation as it enables uniformity of procedures, comparability, and adaptability. When guidelines implement another 1st choice antibiotic, due to for example increasing resistance, the professional companies can easily adapt this in their data processing procedure.

Indicative for the reliability of the procedure and QIs were the as to be expected inter-QI correlations: highly significant correlations were found between high overall prescribing and high prescription percentages for upper and lower RTIs, and between overall high 2nd choice prescribing and low 1st choice prescribing for tonsillitis and pneumonia. Moreover, interesting correlations were seen for QIs that are not necessarily linked, like high overall 2nd choice prescribing with prescribing percentages for RTIs, and with high overall prescribing. This implies that practices have an overall (numbers, prescribing percentages and choice) good, or less favorable performance. The detailed feedback highlights specific targets for improvement for the latter group. GPs’ appreciation of the procedure and QI set will aid in motivating them to appropriate prescribing practice [18].

Some limitations should be acknowledged, most of which were also mentioned by the GPs in the evaluation. First, GPs from group practices expressed a need for outcomes at the individual GP level. Unfortunately, data capture per individual GP is complicated from most electronic patient systems, and even more difficult in the procedure using episodes. Moreover, determining the patient population belonging to an individual GP in a group practice is complicated. Second, in some electronic patient systems, chronic or preventive use of antibiotics is registered as a weekly prescription. Those 52 prescriptions per year disproportionally add in QI1. In refining the set, focus will be on omitting these prescriptions from QI1. Third, for reliability of the procedure, patient contacts need to be ICPC-coded and correctly coded. Correct ICPC coding is stimulated in the Netherlands, but can still be improved. The variation in (correct) ICPC coding is reflected by the variation in the percentage of prescriptions captured in the episodes. We estimated that around 50–60% of all prescriptions would be for an RTI or cystitis in women [20], and a mean of 40% shows that episodes could have been missed. It was, however, reassuring that none of the QI outcomes correlated with this percentage. Fourth, GPs often suggested to correct the outcomes for particular features of their patients (comorbidity, age, social economical state, antibiotic allergies). Beforehand it was decided not to correct for case-mix and individual patient characteristics as many studies have shown that the majority of inter-practice variation in antibiotic prescribing cannot be explained by patient characteristics, and is likely to be due to GPs’ prescribing habits [30,31,32,33]. With respect to the methodology, this pilot set was defined by experts. Feedback from the 45 practicing GPs has been used to further refine and extend the pilot set towards the set that is currently being implemented in the Netherlands. Results from an additional 200 practices will become available soon and show the generalizability of the results.

### 4.4. Perspectives

This proof of concept study has shown that electronic patient systems routinely used in Dutch primary care can reliably deliver outcomes of clinical indication-linked antibiotic prescribing QIs. Thereby, a useful tool has been established to regularly monitor antibiotic prescribing quality per GP practice, for internal use by the GPs, and for external use by the government, the Dutch college of GPs, and educational organizations. Herewith, the way is paved for the implementation of this procedure to support national antibiotic surveillance and stewardship programs. 

## Figures and Tables

**Table 1 antibiotics-09-00670-t001:** Outcomes of the new set of antibiotic prescribing quality indicators.

Quality Indicator	Mean (SD)	Min–Max	Median	25–75
1 Antibiotics/1000 patients/year	303 (92.7)	129–484	311	232–368
2 % Amoxi/clav	10.1 (2.6)	4.7–14.9	10.2	8–12.2
3 % Macrolides	10.2 (4.2)	3.9–21.7	10.1	6.7–12.5
4 % Quinolones	6.5 (1.9)	3.2–12.2	6.4	5.1–7.7
5 % Amoxi/clav + Macrolides + Quinolones	26.7 (4.9)	16.8–40.4	26.9	22.6–30.1
6 Prescribing % URTI	28 (11.8)	8.5–52.6	23.4	19.4–38.1
7 Prescribing % LRTI	33.5 (14.4)	10.5–64.4	29.6	22–44.1
8 % 1st Choice prescribing tonsillitis	60.5 (23)	0–100	62.5	50–75
9 % 1st Choice prescribing pneumonia	74.4 (12.9)	38.5–100	75.6	65.8–84.7
10 % 1st Choice prescribing cystitis (♀)	83.6 (5.9)	60.9–91.7	84.9	80.6–87.2

Mean, standard deviation (SD), the lowest and highest values (Min–Max), median and quartiles (25–75) are shown for the 10 QI outcomes for 44 GP practices.

**Table 2 antibiotics-09-00670-t002:** Correlations between the quality indicator outcomes.

QI	AB	%A/c	%M	%Q	%SUM	%uRTI	%lRTI	%1st T	%1st P	%1st C	% P_E
**Antibiotics/1000 pnt/year**		x	x	x	0.509 <0.001	0.713 <0.001	0.808 <0.001	x	x	x	x
**% Amoxi/clav**			x	x	0.557 <0.001	0.451 0.002	0.383 0.01	x	−0.444 0.003	x	x
**% Macrolides**				−0.314 0.038	0.757 <0.001	0.44 0.003	0.437 0.003	−0.521 <0.001	x	x	x
**% Quinolones**					x	x	x	x	x	x	x
**% Amox/clav + Macrolides + Quinolones**						0.611 <0.001	0.609 <0.001	−0.357 0.018	−0.397 0.008	x	x
**% Upper RTI**							0.824 <0.001	x	−0.398 0.007	x	x
**% Lower RTI**								x	−0.343 0.023	x	x
**% 1st Choice** **Tonsillitis**									x	x	x
**% 1st Choice Pneumonia**										x	x
**% 1st Choice Cystitis (♀)**											x

Pearson’s correlations between the 10 QI outcomes are shown, and also with the percentage of prescriptions captured in the episodes (P_E). A strong positive association is shown in dark green and a weaker association in light green. A strong negative association is shown in red and a weaker association in orange. Correlation coefficients and *p*-values are shown in the cells.

**Table 3 antibiotics-09-00670-t003:** General practitioners’ opinions of the new set of antibiotic prescribing quality indicators.

Questionnaire Item	Ranges of Answers	% of GPs Responding with Options 5–7 * or Yes ^#^
Need for new antibiotic prescribing QIs	no need—very high need *	88%
Added value of QIs linked to clinical indication	yes/no ^#^	100%
Impression of the QIs presented today	not at all useful—very useful *	91%
Expect this QI feedback to change your prescribing behaviour	yes/no ^#^	69%
Recognize yourself in the QI outcomes	not at all—completely *	80%
Trust in data extraction/processing of your routine care data	no trust at all—very high trust *	83%

A total of 45 out of 104 GPs responded to the online, anonymous survey. Free text responses are described in the Results. * Percentages of options 5–7, ^#^ percentages of yes.

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
