# Peer review of "Structural Antibiotic Surveillance and Stewardship via Indication-Linked Quality Indicators: Pilot in Dutch Primary Care"

_antibiotics, 2020, doi:10.3390/antibiotics9100670_

Round 1
Reviewer 1 Report
General:
the authors should follow the author guidelines regarding the formatting of the paper.
L42: Please include the following reference:
Molecules 2019, 24(5), 892; https://doi.org/10.3390/molecules24050892
L45: Please include the following reference:
Antibiotics 2020, 9(2), 97; https://doi.org/10.3390/antibiotics9020097
L46: 80% in human medicine, right? because overall, veterinarians should also be considered!
L49-54: the authors should also include a sentence regarding that social studies and phenomenological studies are emerging, as they provide important underlying information on the causes of imprudent AB use.
Methods:
L78: please elaborate a bit more on the consensus method
L134: what are the exact possible choice on the 7-point Likert-scale
L156: please provide a reference for this statement
Results: appropriate
Discussion:
the authors should elaborate more on the main findings of the study, in addition, compare your results with similar reports from the literature.
Author Response
Reviewer 1: Thank you for your careful evaluation of our manuscript. Find our responses and revisions to your comments below.
General: the authors should follow the author guidelines regarding the formatting of the paper.
We have adapted the style of the abstract to the format of Antibiotics. I presume that the editorial office will do the remaining of the formatting as on previous occasions?
L42: Please include the following reference: Molecules 2019, 24(5), 892; https://doi.org/10.3390/molecules24050892.
Thank you for this suggestion, we have included this reference in line 45.
L45: Please include the following reference: Antibiotics 2020, 9(2), 97; https://doi.org/10.3390/antibiotics9020097.
Although self-medication of antibiotics is a relevant topic and needs to be addressed in many countries, in the context of our study self-medication is less relevant as OTC selling of antibiotics is forbidden.
L46: 80% in human medicine, right? because overall, veterinarians should also be considered!
For clarification, we’ve added: in the outpatient setting in line 42.
L49-54: the authors should also include a sentence regarding that social studies and phenomenological studies are emerging, as they provide important underlying information on the causes of imprudent AB use.
We have added a few sentences describing causes of inappropriate antibiotic prescribing identified by qualitative research studies, in lines 47-52.
L78: please elaborate a bit more on the consensus method
The method we used for defining the set of QIs is described in more detail now in lines 91-99. We removed the reference for ‘focus group discussion’ as we discussed and defined the QIs in a group of experts, without guidance by a moderator.
L134: what are the exact possible choice on the 7-point Likert-scale
Additional text is added also referring to the Table: Answers were allowed on a 7-point Likert scale with answer options ranging from ‘not at all …’ to ‘very …’ and with answer option 4 as ‘moderate’ in lines 161-164. The specific wording of the questions and answer options are shown in Table 3 in the Results.
L156: please provide a reference for this statement
For clarity ‘In the Netherlands’ was added, and this statement has two references: the treatment guidelines of the Dutch college of GPs and a study analysing antibiotic prescribing for all clinical indications in the Netherlands, lines 184-187.
Discussion: the authors should elaborate more on the main findings of the study, in addition, compare your results with similar reports from the literature.
We have added substantially to the Discussion section, with more emphasis on the novelty of our procedure and national antibiotic surveillance and stewardship. Comparison with similar studies proved difficult, as we couldn’t identify studies using routine care data and analysing disease episodes instead of individual contacts, which is added in lines 319-320.
Reviewer 2 Report
using administrative dataset for surveillance of antimicrobial use to avoid over or unnecessary prescribing is valuable resource and can be an efficient strategy to have great impact on a population level. This study identified 10 QIs and collected feedback of the experts and the impacted physicians. The readers would likely want to know more about how it was done, the process of consensus, and how to adapt the proposed methodology to their own surveillance strategies.
Introduction
- to introduce briefly methods used (i.e., what method is being used to create the QIs and the reason to choose such method, use of online survey and its purpose)
Methods
- Expand on the methodology used in building consensus, such as:
- number of experts on the panel, how panel was created
- step used to build consensus
- if there was any step taken to avoid bias and attrition due to fatigue
- definition of consensus (e.g., 100% in agreement; if not 100%, how it was dealt with)
- How were the indicators chosen for consensus building/discussion
- What clinical conditions were included and rationale for only including those conditions
- Add information on the GPs that received feedback - who are they, how were they chosen, how did the GPs receive antibiotic prescribing feedback
Online questionnaires
- authors indicated "the day after the meeting" - please clarify what meeting was this
- how were the survey sent
Results
- online survey response rate
- Table 3 - please define the header more clearly so the data can be easily understood
Discussion
- to further discuss results found (e.g., Table 2 the correlation between the QIs) and its significance
- to compare the 10 QIs in this manuscript with current literatures. Have they been used by others?
- can the results be generalizable to the rest of GP practices based on the output of the consensus panel selection and the voluntary participating GPs
- strength and limitation of the consensus building method and online survey
Author Response
Reviewer 2: Thank you for your careful evaluation of our manuscript. Find our responses and revisions to your comments below.
Introduction: to introduce briefly methods used (i.e., what method is being used to create the QIs and the reason to choose such method, use of online survey and its purpose)
The method used for the main aim of the study is briefly described in the Introduction now, lines 82-85.
Methods: expand on the methodology used in building consensus, such as: number of experts on the panel, how panel was created, step used to build consensus, if there was any step taken to avoid bias and attrition due to fatigue, definition of consensus (e.g., 100% in agreement; if not 100%, how it was dealt with)
The method we used for defining the set of QIs is described in more detail now in lines 91-99, including number of experts, composition of the panel, and that complete consensus was reached for each QI. No steps were taken to avoid bias. Moreover, we removed the reference for ‘focus group discussion’ as we discussed and defined the QIs in a group of experts, without guidance by a moderator.
Methods: How were the indicators chosen for consensus building/discussion. What clinical conditions were included and rationale for only including those conditions
The Methods section now describes the four elements the discussions were based on, lines 93-96. Considerations of the expert group in choosing the specific clinical conditions are described in more detail now in the Results section, in lines 203-206 and in lines 219-225.
Methods: Add information on the GPs that received feedback - who are they, how were they chosen, how did the GPs receive antibiotic prescribing feedback
This aspect is described in more detail now in the Methods under Participants, in lines 149-156.
Methods: Online questionnaires: authors indicated "the day after the meeting" - please clarify what meeting was this, how were the survey sent
These aspects are described in more detail now in the Methods in lines 149-156, and in lines 159-160.
Results: online survey response rate
The response rate to the online survey is mentioned in the legend of Table 3 now, line 290.
Results: Table 3 - please define the header more clearly so the data can be easily understood
The header and footer of Table 3 have been rewritten.
Discussion: to further discuss results found (e.g., Table 2 the correlation between the QIs) and its significance
We have added substantially to the Discussion section. The relevance of the correlations between the QIs is better explained now in lines 352-359.
Discussion: to compare the 10 QIs in this manuscript with current literatures. Have they been used by others?
Our manuscript describes a new procedure and set of antibiotic prescribing QIs. Comparison with similar studies proved difficult, as we couldn’t find studies using routine care data and analysing disease episodes instead of individual contacts. This has been added in lines 319-320. The same set with an additional 4 QIs is being implemented now in the Netherlands and results for over 200 practices will be available soon.
Discussion: can the results be generalizable to the rest of GP practices based on the output of the consensus panel selection and the voluntary participating GPs
In this proof of concept study we’ve analysed outcomes for 44 practices. Currently running studies will show whether the results of these first practices are generalizable, which is added in lines 384-385.
Discussion: strength and limitation of the consensus building method and online survey
We have added that five experts were involved in the definition of the QIs, and that feedback from 45 practicing GPs has been used to refine and extend the set, lines 381-384.
Reviewer 3 Report
The authors describe the development and acceptance of QI metrics for ambulatory prescribing in general practitioner offices. This is an interesting piece, I think outside of the Dutch cohort as well. I have a few minor comments. Thank you!
General:
There is some casual language throughout and opportunities to improve word flow, but these can be corrected during revision. Please re-read carefully. (e.g. opening line of background)
Introduction:
What is the rate of AMR in the community in the Netherlands? (especially among RT isolates, e.g. S. pneumoniae). This will help give context to the 1st line recommendations discussed later.
Stewardship, specifically outpatient stewardship, is not really mentioned. To increase applicability globally, I would introduce this concept more here.
Methods
Do these GP practices represent a geographic spread across the Netherlands?
Did a registered patient equate to a visit at all, or just registered as a patient?
Results
To clarify, these non-1st choice antibiotics were not linked to an indication but were considered inappropriate regardless?
GPs seemed to be favorable. Please clarify the "Response" column in the table 3 and what the % represents.
Discussion
I think this is another great opportunity to circle back to ambulatory stewardship, again increasing broad application.
Another limitation is the assumption of these non-1st line without knowledge of prevalence of things like resistance rates, cultures obtained, rapid diagnostics, allergies, etc. It would be helpful to briefly contectualize some of these for the reader.
Author Response
Reviewer 3: Thank you for your careful evaluation of our manuscript. Find our responses and revisions to your comments below.
General: there is some casual language throughout and opportunities to improve word flow, but these can be corrected during revision. Please re-read carefully. (e.g. opening line of background)
We have carefully re-read the paper and hope to have had corrected casual language.
Introduction: what is the rate of AMR in the community in the Netherlands? (especially among RT isolates, e.g. S. pneumoniae). This will help give context to the 1st line recommendations discussed later.
We have added this highly relevant issue in the Results section describing the considerations in defining the QIs, in lines 229-236.
Introduction: stewardship, specifically outpatient stewardship, is not really mentioned. To increase applicability globally, I would introduce this concept more here.
Thank you for this valuable suggestion, outpatient antibiotic stewardship is now mentioned in the Introduction in lines 54-59, and referred to in the Discussion.
Methods: Do these GP practices represent a geographic spread across the Netherlands?
The geographic spread across the Netherlands is described now in the Methods in lines 155-156. In the next phase of this implementation process outcomes of a refined and extended set were recently determined for 200 practices from all over the Netherlands.
Methods: Did a registered patient equate to a visit at all, or just registered as a patient?
This number equates to all registered patients, irrespective of any visit, which is added now in line 179.
Results: To clarify, these non-1st choice antibiotics were not linked to an indication but were considered inappropriate regardless?
QI2-5 were not linked to clinical indications. We completely agree that part of these were prescribed with valid reasons: allergy, complicated disease, STDs. This percentage is nevertheless considered informative as the higher this percentage is, the more likely that these antibiotics are prescribed inappropriately too. We’ve clarified this now in lines 189-194.
Results: GPs seemed to be favorable. Please clarify the "Response" column in the table 3 and what the % represents.
Clarification has been added to Table 3.
Discussion: I think this is another great opportunity to circle back to ambulatory stewardship, again increasing broad application.
We’ve added this aspect in the Discussion.
Discussion: Another limitation is the assumption of these non-1st line without knowledge of prevalence of things like resistance rates, cultures obtained, rapid diagnostics, allergies, etc. It would be helpful to briefly contextualize some of these for the reader.
Some of these aspects are covered now in lines 189-194, 376-381 and 347-351, and we furthermore assume that the prevalence of allergies and resistance rates are pretty comparable for Dutch patient populations.